# Heterogeneous networks: Fair power allocation in LTE-A uplink scenarios

**Reben Kurda** ⊙ *

Department of Information System Engineering Techniques, Erbil Technical Engineering College, Erbil Polytechnic University, Erbil, Iraq

* reben.kurda@epu.edu.iq

**Data Availability Statement:** All relevant data are within the paper.

**Funding:** The author received no specific funding for this work.

## Abstract

Effective management of radio resources and service quality assurance are two of the essential aspects to furnish high-quality service in Long Term Evolution (LTE) networks. Despite the base station involving several ingenious scheduling schemes for resource allocation, the intended outcome might be influenced by the interference, especially in heterogeneous scenarios, where many kinds of small cells can be deployed under the coverage of macrocell area. To develop the network of small cells, it is essential to take into account such boundaries, in particular, mobility, interference and resources scheduling a strategy which assist getting a higher spectral efficiency in anticipate small cells. Another challenge with small cellular network deployment is further analyzing the impact of power control techniques in the uplink direction for the network performance. With that being said, this article investigates the problem of interference in LTE-advanced heterogeneous networks. The proposed scheme allows mitigation inter-cell interference through fractional self-powered control performed at each femtocell user. This study analyzes a scheme with optimum power value that provides a compromise between the served uplink signal within unwanted interference plus noise ratio to enhance spectral efficiency in terms of throughput. In particular, the maximum transmit power for user equipment in uplink direction should be reviewed for small cells as a major contributor to the interference. The simulation results showed that the proposed fractional power control approach can outperform the traditional power control employed as a full compensation mode in small cell uplinks.

## 1. Introduction

Mobile telecommunications technologies such as General Packet Radio Service (GPRS) and Universal Mobile telecommunications System (UMTS) have mainly focused on quality of services (QoS) of non-Real Time (RT) and voice applications. Nowadays, the development of real applications for cellular devices is mostly concentrated on multimedia services. Accordingly, Long Term Evolution advanced (LTE-A) technology has been introduced by the third Generation Partnership Project (3GPP) to improve capacities of the modern cellular networks and fulfill RT applications services. Hence, the advanced versions of Long Term Evolution (LTE) employ Single Carrier Orthogonal Frequency Division Multiple Access (SC-ODFMA) technology in the Uplink (UL) direction.

**Competing interests:** The authors have declared that no competing interests exist.

The battery life can be considered as one of the key parameters that significantly affect the efficiency of all mobiles. Although battery performance has been improved by many manufactures, it is still necessary to ensure that the mobile uses as low battery power as possible. As mentioned above, LTE-A signal format has been using SC-OFDMA to provide reliable high speed data communications. Similar fact has been reported by Myung, Lim [1]. The use of OFDM has enabled LTE to provide reliable link quality, even in the presence of reflections and the adaptive modulation, which provides the ability to modify the link according to the prevailing signal quality. In other words, QoS of LTE technology should work in a way that optimizes the fairness and utilization. In addition, RT services must have a high QoS level, but non-RT services may only need a minimum bitrate [2, 3]. To increase the capacity in terms of throughput, network densification (number of base station) is necessary for wireless access networks [4]. In other words, the distance between User Equipment (UE) and Traditional Base-Station (eNodeB) can be reduced by using extra base station per unit-area that enables accommodating extra UEs per unit-area, and causing a link-budget enhancement [5, 6].

The deployment of additional base-stations (eNodeB) per any unit area makes shorter distances between classical eNodeB and their UE, as well as permits considering more UEs per specific area, and leading in a higher resource capacity enhancement [5]. Due to high density of UE in some specific areas (e.g. metropolitan and urban), the demand to widen the capacity of network has been continuously increasing. Thus, it is important to find an optimum location and best configuration for the network-settings in order to guarantee minimum interference-generation, satisfactory data-rates and better QoS for each UE.

Recently, LTE specifications have optimized eNodeB by deploying a smaller coverage in order to remarkably enhance QoS of future generation wireless technologies such as Femtocell, also called Home eNodeB (HeNodeB). However, to configure the setting of traditional cellular base-station with newly low powered radio access nodes HeNodeBs in any location such as offices, apartments, malls, train stations or houses, it is required to analyse the impact of reusing the licensed spectrum within various attributes, i.e. size of cell, UEs distances among each other, HeNodeB serving, power control parameters (e.g. UL direction), the distance between inter-site neighboring small cells, and also considering the scenario composition [7, 8].

Normally, the main goal of Femtocells is to offer strengthened mobile coverage of UEs in buildings. Accordingly, the issue related to the loss of signals due to multiple wall penetration will be overcame [9].

One of the principal issues of the femtocell utilization, namely closed-subscriber group (CSG), is mainly concentrated on femto-femto and macro-femto interferences. However, this may not happen for the open subscriber group (OSG) mechanisms because the macro-users are permitted to be automatically handed-over [10–12].

To guarantee power efficient utility, minimize the intra-cell interference and enhance QoS, 3GPP has introduced an approach for Transmitting Power Control (TPC) which also authorizes to investigate power controls in both DL and UL directions [13, 14].

According to the LTE-A specification (3GPP Release NO. 11) made by Schwarz and Rohde [15] which is titled "LTE- Advanced (3GPP Rel.11) Technology Introduction—White Paper", the coordination multi-point operation (CoMP) has standardized for the Uplink HetNets based on the interference scenario for both resources scheduling and Beamforming (Power allocation). Moreover, specific control techniques for the UE transmission power are addressed to minimize the negative effect of the interference on the network capacities. Furthermore, there were 2 mechanisms introduced for Resources Block (RB) TPC in the LTE uplink: (i) Open Loop Power Control Point (OLPC) and (ii) Dynamic Power Offset (DPO) that provide the basic operating point for UE transmit power (TP) as shown below:

i. OLPC has been introduced as an ability of UE to determine the uplink transmits power to a specific value to fit the receiver. The mentioned process works by assessing the pathloss portion between the eNodeB serving and UE, knowing cell reference and received power of the reference signal, as well as UE-Specific portion which is an offset that applies to a particular UE.

ii. DPO scheme works according to the signaled eNodeB. In addition, DPO has the ability of UE to adapt the uplink transmits power based on the dynamic-loop correction value (TPC commands). In general, eNodeB sends the commands of TPC to UE according to the Signal-to-Interference-Plus-Noise-Ratio (SINR) and MR Signal-to-Interference in order to normalize with the different modulation and coding schemes (MCS) [16–18].

In general, the problematic of interference in uplink and downlink are different subjects. In Downlink direction, higher performances to macro cell user equipment will be the main target [9, 11, 19–23]. While, in uplink direction, higher performances to small cell user equipment will be the main target. Since the domain of this study relates to the uplink direction, the author of this study investigated a model to get higher performance to small cell user equipment. Accordingly, to overcome the mentioned issues (i and ii), this study utilizes a novel strategy for the DPO scheme. In this strategy, the uplink receiver at HeNodeB estimates the SINR of the received signal and compares it with the desired SINR target value according to Real Time Traffics (RTT). Based on the mentioned parameters, this study named the strategy "Fractional Self-Power Control (FSPC)". In the proposed strategy, TPC command is transmitted to UE to increase the power transmitter power when the received SINR does not satisfy the SINR target. Otherwise, the TPC command will be requested to decrease a transmitter power. In other words, the FSPC specifies the minimum transmission for each UE in the first step. Then, the initial power will be further adjusted using PDCCH signaling as a TPC commands to the UE.

The remainder of this study is structured seven sections. Section 2 describes the related studies. For that purpose, an extensive literature has been devoted regarding the LTE. Section 3 describes the contribution of the proposed strategy (FSPC) to the scientific community. Then, the System Model and Assumptions are introduced in section 4. Section 5 discusses the proposed scheme. In section 6, the simulation environment scenarios within the numerical results are described. Finally, the output and conclusions of this study are shown in Section 7.

## 2. Background

In general, the DPC scheme is adopted for eNodeB deployment, which guarantees a higher QoS for the average of total cell's users by slightly minimizing the performance of the edge cells [24]. Previous studies [25–27] investigated the influence of ULTP in detail. Nevertheless, some assumptions regarding stability of channels over different frames, namely in mobility cases, were not taken into consideration.

Another study [28] also showed the reduction of the transmission energy in regard to the uplink allocation resources. However, the study evaluated the gain with the conventional power consumption without eNodeB interference. On the other hand, concerning the TPC mode, many research studies investigated the small cell deployments. In a study [29], bandwidth allocation issue and DLTP for a HeNodeB in heterogeneous network scenario have been explored in details.

In other studies [30, 31], a decentralized power controlling scheme with a fully compensation mode for small cells in each homogenous and heterogeneous network deployment scenarios were proposed. Although the small cell UL deployment was investigated, the lower

transmission power might be differently expected. On the other hand, to obtain higher insight concerning the influence of the maximum transmission power, it is important to analyse its influence on HeNodeB and eNodeB cell functioning and the TPC mode by considering OLPC.

In some studies [32–36], the decentralized power allocation schemes were proposed under the assumption of limited eNobeB user channel information. In the neighbor-friendly scheme of Kim, Je [33], HeNobeBs allocated the total adjusted power on sub-channels in order to maintain the targeted Femtocell User Equipment (FUEs) sum-rates while mitigating the interference on Macrocell User Equipment (MUEs). Nevertheless, some assumptions about channel stability over several frames may not be valid, especially in mobility scenarios.

Bennis, Giupponi [34] analyzed the influence of various eNobeB sizes on the PC, assuming femtocells' overhearing ability of MUEs' RSRP measurements. DL power allocation methods were proposed in some studies [35, 36], where no control information exchange was considered between HeNobeBs and eNobeBs. As an alternative, each HeNobeBs predicts the RB allocation to its close-by MUEs according to their overheard Channel State Information (CSI) feedback from MUEs to eNobeBs. According to the MUE allocation prediction, each HeNobeBs can perform resource allocation to FUEs while mitigating interference towards close-by MUEs. Nevertheless, interference mitigation (IM) among FBSs was not considered.

Milner, Davis [37] proposed a new Flocking Algorithm for self-organization and control of the backbone nodes in the hierarchical heterogeneous wireless networks through collecting local information from end users. Additionally, the new algorithm suggested by the authors can sufficiently work for the energy optimization while respect to convergence speed and quality of solutions.

To optimize the problem of energy efficiency in the resources allocation in a heterogeneous network scenario, Zhang, Liu [38] proposed a new power and sub-channel algorithm allocations according to the number of the users and their demands. Accordingly, they proposed to use classical small/macro cell case networks instead of the conventional systems. Similar concept can be seen in the study of Chen, Saad [39] regarding the optimization of the UL–DL decoupling use, and Iturralde, Yahiya [40] for radio resource management schemes. Furthermore, Tsiropoulou, Kapoukakis [41] focused on the optimum range of power allocation and subcarrier in multiuser SC-FDMA wireless networks through the tradeoff between users' actual uplink throughput performance and their corresponding energy consumption. Nevertheless, this work did not focus on heterogeneous network.

Moreover, Tsiropoulou, Kapoukakis [42] made an extensive survey to resource allocation algorithms in multiuser SC-FDMA wireless network through throughput optimization, resource allocation and fairness, resource allocation considering user's Qos requirements, and joint power and subcarrier allocation. Accordingly, the authors reported several promising paths to follow heterogeneity network design and implement a resource allocation scheme such as subcarriers/RBs and users' uplink transmission power with respect to multiple services (VoIP and video flow) which is related to the scope and novelty of this study.

Regarding the performance of small/macro cell, a few studies [43, 44] have focused on the output of different small cell stationing scenarios. Jensen, Lauridsen [45] reported that DL and UL TP data rate significantly influence the power consummation of UE. Thus, it is important to focus on the influence of maximum power limit for any LTE UL to enhance the battery life of UE. Joshi, Colombi [46] focused on the radio frequency electromagnetic field (RF-EMF) and suggested that the UE TP must be kept lower to minimize the emissions.

Recently, with continuous increase of user requirements, the capacity and performance have been improved. Consequently, energy-efficiency in cellular networks based on the dynamic power allocation has taken remarkable attention [47]. The aim of considering energy-efficient communication is to reduce the total network energy consumption and achieve an acceptable rate of data transfer of the user [48]. To achieve the mentioned goal,

there are two main methods to use small cell base station: (i) static (fixed) [49, 50] and (ii) dynamic (movable) [51–54] small cell base stations deployment.

Additionally, the dual connectivity-based seamless HO procedure was introduced in a study by Huang, Tang [54] to guarantee the transmission QoS of UEs. The authors proposed an approach of transferring UEs from small cell to macrocell base station. Also the approach reduced the network energy consumption and mitigated the inter-cell interference. The main goal was to enhance the energy-efficiency in the DL, however, the energy-efficiency enhancement for the UE in UL direction was not taken into consideration in the mentioned study.

To enhance network accessibility in urgent situations, Teng, Li [55] focused on the efficient usage of autonomous mobile base-stations (vehicular or aerial). This was mainly used to overcome the issue of disconnected network. Similarly to the mentioned study, Cileo, Sharma [56] also focused on the aerial base station and made a broad analysis on its inter-cell interference, capacity and cell-coverage.

Although the dynamic base stations deployment may solve some issues (e.g. discounted networks and reduce the impact on the main stations in loaded areas), there are some disadvantages due to the fact that UE may find frequencies from multiple stations that results to consume more power. Some researchers [57–59] have studied the mentioned issue of the dynamic base stations and reported some ways to overcome the obstacles.

Apart from the mentioned studies [57–62], other researchers have also investigated the IM. For example, López-Pérez, Valcarce [63] showed some instructions to solve the interference-mitigation and spectrum-allocation issues. Blough, Resta [64] emphasized the significance of SINR based-model for IM. Uygungelen, Auer [65] proposed an approach to solve the interference-mitigation by splitting the main frequency band into sub-bands. Madan, Sampath [66] proposed another scheme by considering delay based and best effort flows. Huang, Berry [67] suggested another approach to solve interference-mitigation by utilizing a game theory scheme.

According to the literature review shown above, regarding the proposed approaches to solve the interference-mitigation, no method can be found for the power control to minimize femtocell-interference over LTE in guarantee of RT services requirements. Therefore, in this contribution, the network-performance is evaluated to clarify the requirements for a different Power Max limit for a small cell UL direction in each of heterogeneous network cell center and cell edge, in accordance to the 3GPP potential that identifies the maximum transmit power for the micro cell (Pmax = 23 dBm) but it does not specify for small cells.

## 3. Research significance

The distinctive contributions of this work are on introducing a strategy which holds an effective mechanism to mitigate interference in HeNodeB scenario by using DPO transmission control called FSPC. The innovations of this paper are summarized on the following aspects (i-v). In other words, they can be considered as a main contribution of the study.

i. Outlining two small cell deployment scenarios within the existing macro layer which were:

  - Heterogeneous network deployment with small cells near macro cell eNodeB;

  - Heterogeneous network deployment with small cells near macro cell edge.

ii. Due to the derivation for interference, received SINR and spectral efficiency proportionally, the proposed strategy for heterogeneous network deployment scenarios carries out a constant haggle between throughput and interference the by TCP transmission power value.

iii. Numerical computation of reduced *P*max limit for each scenario for the small cell uplink.

iv. Since the SC-OFDMA technology is mainly introduced to transfer data in LTE-A UL direction, the proposed strategy simulated the RT services by the same technology (SC-OFDMA) with a scenario that considers VoIP flows and Video.

v. This method can be considered an applicable strategy because it overcomes the interference issues in two respective conditions (i) targeting minimum threshold for the RT services and (ii) then applying boosting factor that enables adapting the power allocation proportionally to the femtocells/Macrocell' role in the interference situation rather than uniformly as usually performed in traditional power adjustment approaches. Thus, the proposed strategy can be easily used by any mobile developer companies.

## 4. System model and assumptions

### 4.1. System model

This study considered a eNodeB/HeNodeB overlay system where $L$ HeNodeB are uniformly distributed in one eNodeB $M$ coverage area in two ring of distance and neighboring HeNodeB were considered around eNodeB, the different scenario cell-center and cell-edge considered, see Figs 1 and 2 shows the heterogeneous network and the distribution of HeNodeB at each of the of the mentioned area, respectively.

This study assumed that $I$ MUEs are uniformly distributed within the eNodeB coverage area and are the non-CSG to all HeNodeB $F_l$, ($l = 1;...;L$) when they approach its coverage area. To perform the resource allocation of their RBs on the UL direction, both eNodeB $M$ and HeNodeB $F_l$ require the received signal quality on the UL direction expressed by SINR before

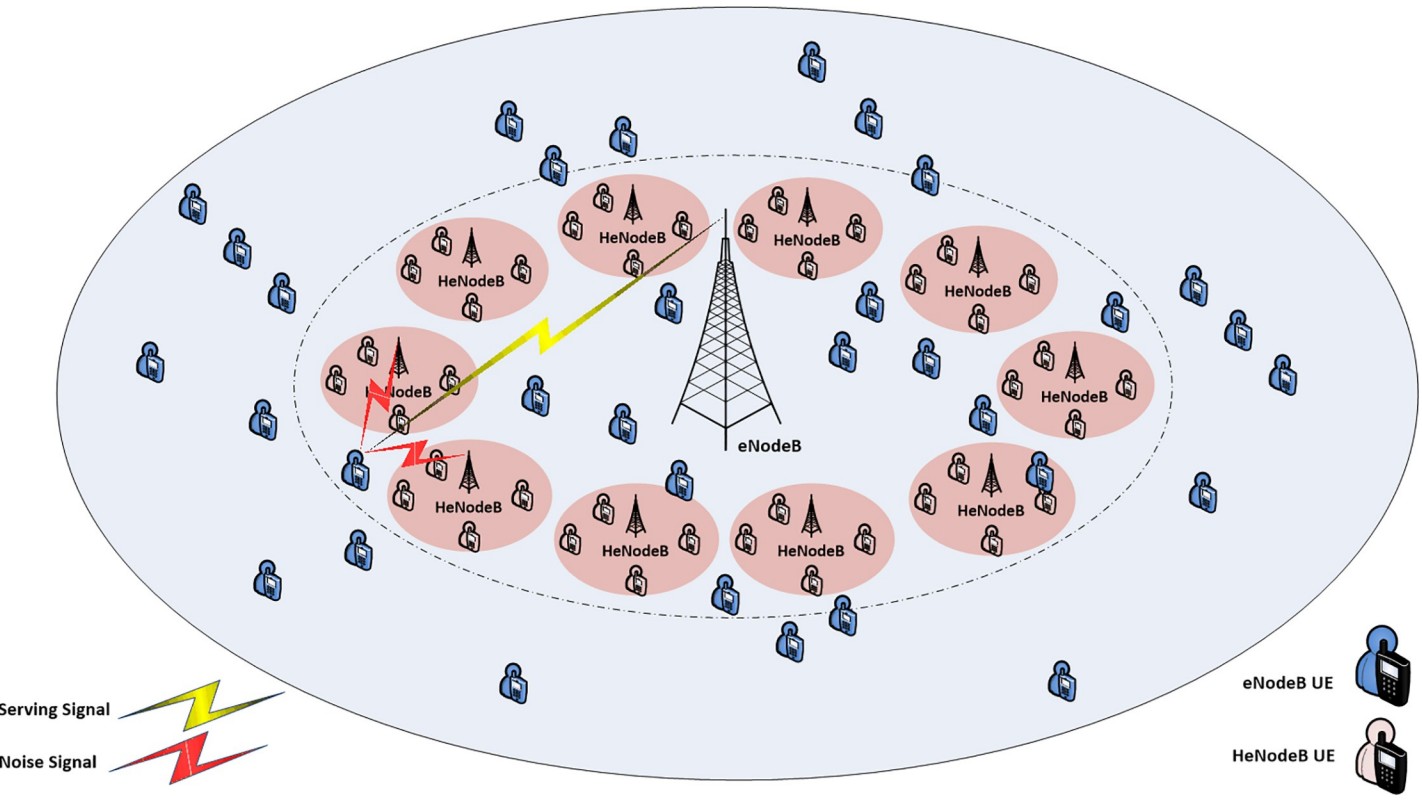

**Fig 1. Cell center area of the macro cell.**

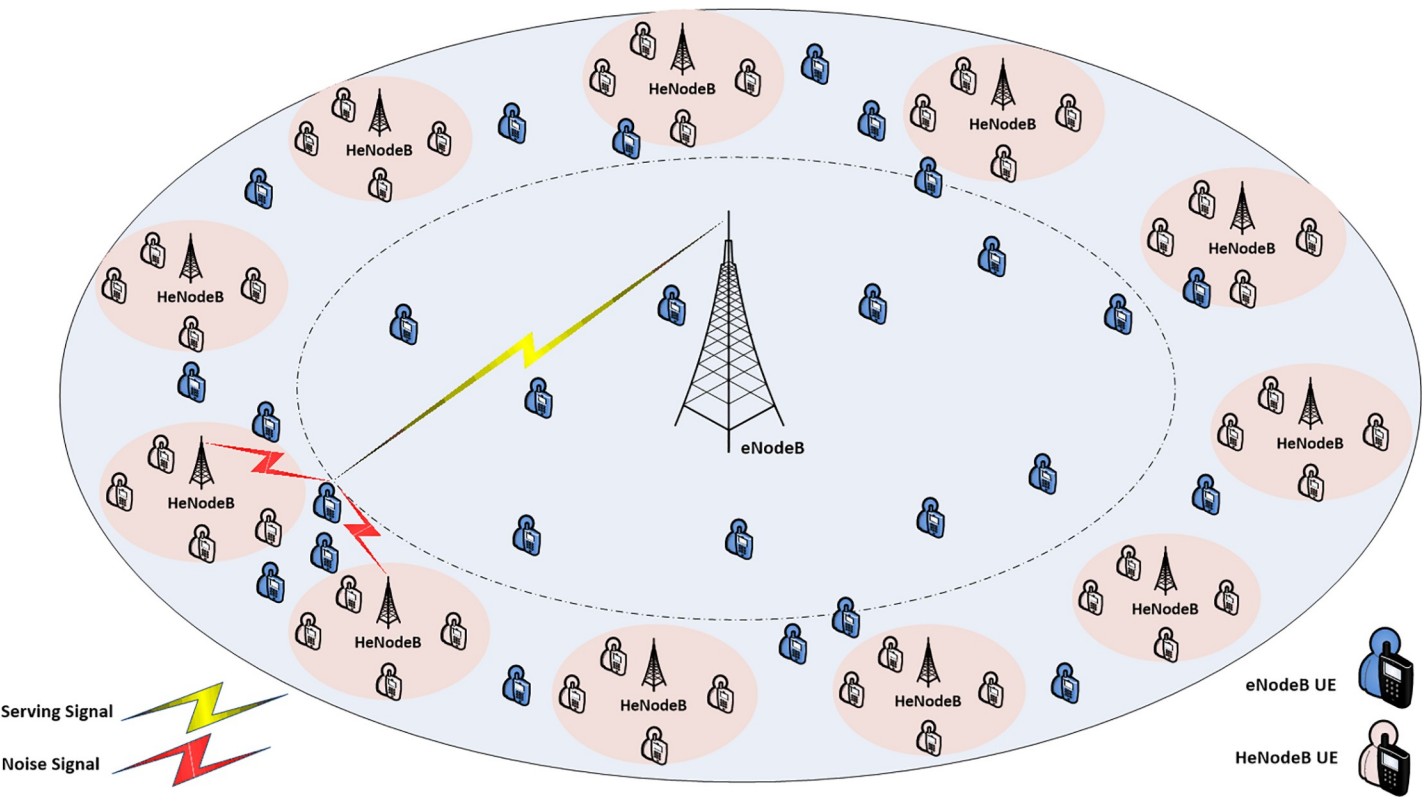

**Fig 2. Cell edge area of the macro cell.**

any modulation and scheduling grant to UE. The received SINR of a given eNodeB $M$ granted to a $MUE_i$ on RB $k$ ($k = 1;...;K$) can be expressed as Eq 1. Similarly, the UL $SINR^k_{FUE_j,F_l}$ of a given $FUE_j$ associated with a given femtocell $F_l$ on RB $k$ can be expressed as Eq 2.

| Eq 1 | Eq 2 |
|---|---|
| $$SINR^k_{MUE_i,M} = \frac{\frac{P^k_{MUE_i}}{PL^k_{MUE_i,M}}\lvert G^k_{MUE_i,M}\rvert^2}{\sum_{FUE_j \in Z_{i,k}} \frac{P^k_{FUE_j}}{PL^k_{FUE_j,M}}\lvert G^k_{FUE_j,M}\rvert^2 + N_0}$$ | $$SINR^k_{FUE_j,F_l} = \frac{\frac{P^k_{FUE_j}}{PL^k_{FUE_j,F_l}}\lvert G^k_{FUE_j,F_l}\rvert^2}{N_0 + A + \sum FUE_m \in V_{j,k} \frac{P^k_{FUE_m}}{PL^k_{FUE_m,F_l}}\lvert G^k_{FUE_m,F_l}\rvert^2}$$ $$m \neq l$$ |
| Where $P^k_{MUE_i}$ the current is TP allocated on RB $k$ to the serving cell eNodeB $M$, and $\lvert G^k_{MUE_i,M}\rvert^2$ is the channel fading gain between $MUE_i$ and eNodeB M on RB $k$. Similarly, $P^k_{FUE_j}$ is the TP of neighboring HeNodeB user $FUE_j$ on RB $k$, while $Z_{i,k}$ represents the set of all interfering $FUEs$ on user $MUE_i$ on RB $k$ toward the UL direction. Similarly, $\lvert G^k_{FUE_j,M}\rvert^2$ is the channel fading gain between $FUE_j$ and the neighboring eNodeB $M$ on RB $k$. $PL^k_{MUE_i,M}$ and $PL^k_{FUE_j,M}$ represent the pathloss from $MUE_i$'s to its serving cell eNodeB $M$ and all neighboring $FUE_j$ respectively, and $N_0$ is the power of the additive white Gaussian noise. All the MUEs are considered non-Closed Subscriber Group users for $F_l$'s operator. Each $F_l$, serves $J$ FUEs distributed randomly in its coverage area. The transmissions are SC-OFDMA based and all MUEs and FUEs share the same frequency band divided in $k$ RBs. We also assume that eNodeB M and HeNodeB L are fully loaded and cannot execute any intra-channel allocation to mitigate the potential interference caused to FUEs on the UL direction. | where $P^k_{FUE_j}$ is the current TP allocated on RB $k$ by the $FUE_j$, and $\lvert G^k_{FUE_j,F_l}\rvert^2$ is the channel fading gain between $FUE_j$ and its serving HeNodeB $F_l$. $P^k_{FUE_m}$ is the TP of a neighboring $FUE_m$, while $V_{j,k}$ represents the set of all interfering $FUE_m$ on user $FUE_j$ on RB $k$. $\lvert G^k_{FUE_m,F_l}\rvert^2$ is the channel fading gain between $FUE_m$ and its neighboring FBS $F_l$. $PL^k_{F_m,F_l}$ stands for the pathloss between $FUE_m$ and FBS $F_l$. A is the interference generated by $MUE_i$ on RB $k$, namely $A = \frac{P^k_{MUE_i}}{PL^k_{MUE_i,F_l}}\lvert G^k_{MUE_i,F_l}\rvert^2$. |

## 4.2. Assumptions

This study mainly emphasized the TPC approach because of its significance in SINR as described in Eqs 1 and 2. According to the specification of 3GPP, each femtocell service has 10X10m coverage surface, while the interference level at the UL direction dynamically changes in regard to the UEs mobility and their transmission power at the current time. For example, if a $FUE_j$ is at the cell center, a fix transmission power with a highest level does not become an interferer to neighbors $FUE_m$ served HeNodeB. In contrast if a $FUE_j$ served at the cell edge and set their transmission value to the highest level as its preferred by the utilizer, which in order to transmit a better quality of signal to the base station, in this case, the $FUE_j$ becomes a serious interferer to all neighboring UEs, especially when the base stations are in full load and can't avoid the interference. This happens where there are no free channels to change the sun-carriers at the resource allocation process.

Now let's assume, if the HeNodeB owner configured their UEs to transmit power with a reasonable value, the interference will be declined but the expense of lower MCS value, therefore the performances in term throughput will not satisfy some kinds of transmission packets such as to real-time flows VoIP and Video.

Thus, the major benefit of this strategy is concentrated on configuring the transmission power value as a dynamic variable which changes depending on the scenario. As mentioned in earlier context, HeNodeB estimates the CSI and reports its UE through PDCCH. Furthermore, each HeNodeb will choose a rating of transmission power based on the user's position for maintaining the consumed power as low as possible with maintaining the throughput higher than the threshold. In this context a non-cooperative strategy between base stations introduced depending on the assumption that each particular ameliorate their transmit power independently without any conversation between particulars nodes such as HeNodeBs/enodeB. In this case, SINR have been chosen as MR to set mobile transmit power as low as possible as well as guaranteed a high Qos.

## 5. Proposed algorithm

As mentioned above, the main aim of this proposed approach is to determine the exact amount of power experienced from each of the interfered HeNodeB for the adjustment purpose. Accordingly, this study introduced a metaheuristic algorithm that sufficiently generates two power adjustment levels called DPO and FSPC. Furthermore, the schemes offer various levels of awareness of FUEs performance, and as a result, mitigate the negative impacts the MUEs or neighboring FUEs output because of the tradeoff between their connecting requirements in regard to the spectral efficiency and satisfy the RT traffic.

More specially, DPO and FSPC perform various selection mechanisms of the interfered femtocells that are applied to adjust the power and adapt the level of the power adjustment (decrease or increase). Apart from the power adjustment parameters, defining such strategies offers an extra leverage in adjusting the optimum between FUEs and MUEs outputs. Further details regarding the mentioned steps are shown below.

In this scenario, different MUEs and FUEs are arbitrarily positioned within their coverage MBS M and FBS respectively. First, each *FBS l* can estimate and check the channel state indicator in terms RSS from each active $FUE_j$ as well as it can listen to other RSS user equipment from any neighboring *FUEs/MUEs* on the UL direction, periodically sends the RSS from their UE to its serving cells. Then, *FBS l* can evaluate the $SINR^k_{FUE_j,F_l}$ as introduced in Eq 2 that impacted each allocated resource (RB *k*), before *FBS l* performs the resource allocation procedure according to the modulation and coding schemes (See Table 1). In order to note that, the RSS measurement report assists *FBS l* to recognize the source of interference and to confirm if it either happens by neighboring cells or if it is the result of any other influence, i.e. thermal

**Table 1. LTE Modulation and Coding Schemes (MCS).**

| MCS | Modulation | Code Rate | SINR [dbm] |
|---|---|---|---|
| MCS1 | QPSK | 1/12 | -4.63 |
| MCS2 | QPSK | 1/9 | -2.6 |
| MCS3 | QPSK | 1/6 | -0.12 |
| MCS4 | QPSK | 1/3 | 2.26 |
| MCS5 | QPSK | 1/2 | 4.73 |
| MCS6 | QPSK | 3/5 | 7.53 |
| MCS7 | 16QAM | 1/3 | 8.67 |
| MCS8 | 16QAM | 1/2 | 11.32 |
| MCS9 | 16QAM | 3/5 | 14.24 |
| MCS10 | 64QAM | 1/2 | 15.21 |
| MCS11 | 64QAM | 1/2 | 18.63 |
| MCS12 | 64QAM | 3/5 | 21.32 |
| MCS13 | 64QAM | 3/4 | 23.47 |
| MCS14 | 64QAM | 5/6 | 28.49 |
| MCS15 | 64QAM | 11/12 | 34.6 |

noises or fast fading. For that purpose, this study defines the Score Interference ($Total_{In}^{k}$) represented by the interference of each the UL transmission power of all interfering femto/macro user in the REM table as follows:

$$Total_{RSS}^{k} = \sum_{x=0}^{y} RSS_{MUE_i,F_l}^{k} + \sum_{i=0}^{n} RSS_{FUE_j,F_l}^{k} \tag{3}$$

Where $\sum_{x=0}^{y} RSS_{MUE_i,F_l}^{k}$ and $\sum_{i=0}^{n} RSS_{FUE_j,F_l}^{k}$ are the total received signal strength by $F_l$ from each of interfering $MUE_i/FUE_i$ on RB $k$, which are expressed as $RSS_{MUE_i,F_l}^{k} = \frac{P_{MUE_i}^{k}}{PL_{MUE_i,F_l}}$, $RSS_{FUE_j,F_l}^{k} = \frac{P_{FUE_j}^{k}}{PL_{FUE_j,F_l}}$.

The ultimate goal of this approach is to identify interfering macrocell or neighboring femtocell users, and their exposure level on the interference circumstances at the UL direction. Then, *FBS l* checks the received signal strength $RSS_{FUE_j,F_l}^{k}$ from the served user $FUE_j$ and compares it with Eq 3 to determine if the given $RSS_{FUE_j,F_l}^{k}$ is greater than $Total_{RSS}^{k}$ or not. If so, *FBS l* identifies all the $MUE_i/FUE_z$ as an interfering user for victim $FUE_j$. $RSS_{M_x,MUE}^{k}$ stands for signal strength by $F_l$ from its served $FUE_j$ on RB $k$ and expressed as $RSS_{FUE_j,F_l}^{k} = \frac{P_{FUE_j}^{k}}{PL_{FUE_j,F_l}}$

To moderate the interference produced by each interfering users, $F_l$ initiates the first power adjustment decision process as well as $F_l$ observing an outage after decreasing the *UL* transmission power from their $FUE_j$ where the target shall be verify following condition.

$$SINR_{F,FUE_j}^{k} \geq SINR^{target} + \Delta SINR \tag{4}$$

where $SINR^{target}$ is a SINR boundary to ensure an acceptable QoS uses by FSPC as a first power selection by $FUE_j$. $\Delta SINR$ is a positive protection perimeter measured in decibels, utilized with $SINR^{target}$ simultaneously to check if $FUE_j$ faces a weak serving signal.

For simplicity, the interfered MUEs performance and the traffics have not been evaluated and it's out of this context. First, for each victim FUE$_j$ on RB k, FBS l listen the set of femtocells or/and macrocells user responsible for this interference and determines their respective

interference values. Second, FBS l instructs their victim to maximize/minimize the transmission power from $P^k_{FUE_jF_l}$ to $P\prime^k_{FUE_jF_l}$ according to the following procedure.

DPO determines the maximum interference resisted by the victim $FUE_j$ on RB k. In other words, $MI^k_{FUE,F_l}$ is used to measure the highest noise that a user can afford when exposed from the neighboring MUEs.

$$MI^k_{FUE,F_l} = \frac{RSS^k_{FUE,F_l}}{SINR^{Target}} - N_0 \tag{5}$$

where $SINR^{Target}$ reflect to the RT application, according to the MCS (See Table 1)

Next, *FBS l* computes to each victim *FUE_j* the initial value of the lower transmission *power* $P\prime_{FUE_jF_l}k$ on *RB k* as

$$P\prime_{FUE_j,F_l}k = \alpha^k_{F_l} \cdot \beta^k_{F_l} \cdot P^k_{FUE,F_l} \tag{6}$$

Where

$$\beta^k_{F_l} = 1 - \frac{MI^k_{FUEj}}{Total^k_{In}}$$

$P^k_{FUE_j,F_l}$ stands for the maximum transmission power of the victim $FUE^k_j$ in RB k, and $Total^k_{In}$ is the received signal strength from all neighboring Interfering UEs (Femcocells/Macrocells) measured by $F_l$ or $FUE_j$ for the allocated RB k. First, the previous power level $P^k_{FUE_jF_l}$ is adapted by the factor $\beta^k_{F_l}$ that corresponds to the ability per-UEs interference level, where $MI^k_{FUEj}$ standardized by the total of all RSSs of interfering femtocells/Macrocells.

Next, FSPC is apply the power adjusted $P\prime^k_{FUE_j,F_l}$ by the boosting factor $\alpha^k_{F_l}$ whose value is proportional to SINR MR which must verified, $SINR^k_{F_l,FUE_j} \geq SINR^{Target}_{GPA}$. However, $P\prime^k_{FUE_j,F_l}$ dynamically change according to RT traffic demand and the Pathloss of $FUE_j$ on the UL direction.

Using the boosting factor $\alpha^k_{F_l}$ enables adapting the power allocation proportionally to the femtocells/Macrocell' role in the interference situation rather than uniformly as usually performed in traditional power adjustment approaches. However, the amount of power boosting will be limited by a tunable system parameter *x*, $(0 < x \leq 1)$ which represents the maximum allowed power reduction/increase ratio of the nominal power $\beta^k_{F_l} \cdot P^k_{FUE_j,F_l}$, i.e.,

$P\prime_{FUE_j,F_l}k \in [\beta^k_{F_l}P^k_{FUE_j,F_l}(1-x), \beta^k_{F_l}P^k_{FUE_j,F_l}(1+x)]$.

Therefore,

$$\begin{cases} 1 - x < \alpha^k_{F_l} \leq 1 & \text{if } SINR^k_{F_l,FUE_j} \geq SINR^{Target}_{GPA} \\ 1 < \alpha^k_{F_l} \leq \min\left(1 + x, \frac{P^k_{FUE_j,F_l}}{\beta^k_{F_l}P^k_{FUE_j,F_l}}\right) & Otherwise \end{cases}$$

Finally, the selection of the determined interfered FUEs femtocells that are applying to promote factor $\beta^k_{F_l}$ defines the DPO scheme. While, $\alpha^k_{F_l}$ value more adjusted according to represent the FSPC strategy. Thus, these condition variable be useful in determine the set of femtocells user which have the largest and shared effect on the global interference circumstances and hence allow a self-adapted power adjustment proportional to this impact and mitigate

femtocell/macrocell user interference through LTE toward satisfying RT service's requirements. Figs 1 and 2 demonstrate a simple interference scheme in each of cell center and cell edge scenario.

## 6. Simulation environment

### 6.1. Simulation parameters

To analyze the effectiveness of the proposed power allocation schemes, several simulations were applied with two different scenarios using LTE-Sim simulator [68]. The network topology consists of one MBS M and several neighboring FBSs positioned according to reasonable distance from M. For that purpose, two different scenarios, which they are considered as the main contribution of this work, have been implemented as follows:

- First, the heterogeneous network deployment with femtocell deployed near the macro cell eNodeB was examined, named macrocell cell center area (see Fig 1).

- Second, the heterogeneous network deployment with femtocells distributed from the macro cell eNodeB is examined, named macrocell cell edge area (see Fig 2).

Each FBS is placed in a building and serves a maximum of four users registered as Close Subscriber Group (CSG) users moving at a low speed compared to the outdoor users (3km/h vs. 30km/h, respectively). The quantity of femtocells that are relatively close (neighbors) start from one until ten, increasing in a single unit (one) for increasing the interference. Thus, each user at each femtocell utilizes all the femtocell resource blocks. While MUEs are randomly positioned within the macrocell and move according to a random walk mobility model. In this study, the MUEs experiments were not taken into account. LTE-Sim gives a significant support for radio resource allocation in frequency and time domain. Piro, Grieco [68] focused on the time domain and showed that the radio resources can be spread every TTI (every single of them lasting one ms). Additionally, every single of TTI was collected by 2 time slots of 0.5ms, equivalent to fourteenth SC-OFDM symbols; ten consecutive TTIs form the LTE frame. The simulation parameters are defined in (Table 2).

In order to confirm that the scalability, efficiency, and robustness of the proposed framework a large scale set up with a large number of connected nodes was considered. Moreover each simulation scenario from 1–10 femtocell (Table 2) was repeated 30 times. At each time, the Macrocell user was positioned in different places in order to get a data more realistic. Accordingly, this feature implemented by using srand() function. Then, the average data was considered to get the performance of Femtocell users in terms of throughput. In addition, the time flow at each scenario specified by 300s which was gave the gigabyte to data flow to both services VoIP and Video.

**Table 2. Simulation parameters.**

| Parameters | Values | Parameters | Values |
|---|---|---|---|
| Macrocells | 1 | Thermal noise density | -174dBm/Hz |
| Femtocells | 1–10 | Carrier frequency | 3.5 GHz |
| MUEs | 60 | UE noise figure | 2.5 dBm |
| FUEs | 4 | Macrocell radius | 500m |
| Bandwidth | 10MHz | Femtocell radius | 200m |
| Total RBs | 50 RBs | FUEs average speed | 3km/hour |
| FUE Pmax power | 23 dbm | MUEs average speed | 30km/hour |
| MUE Pmax power | 23 dbm | SINR$^{target}$ | RT application |

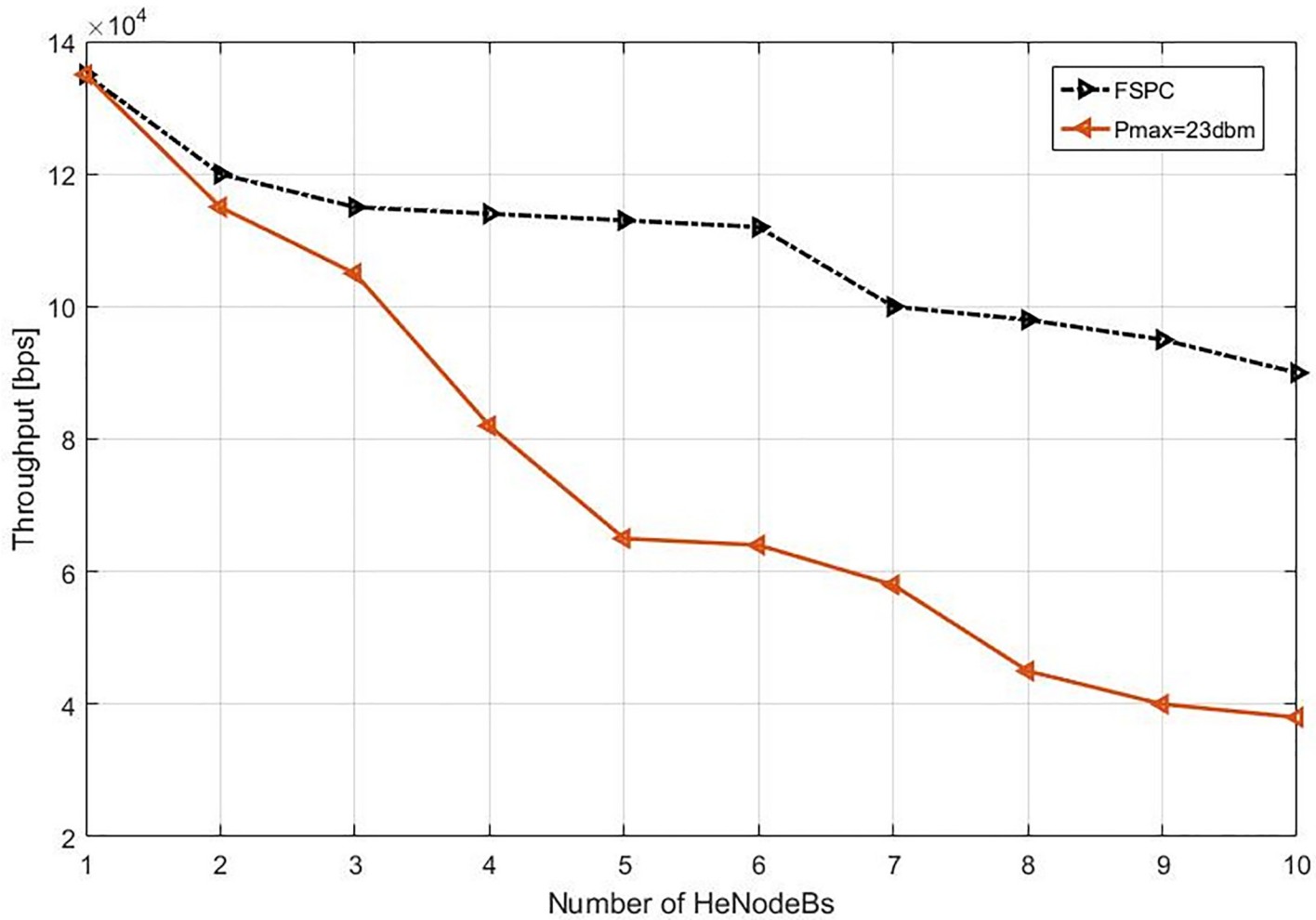

**Fig 3. Throughput average per video flow (cell center).**

This study used a path loss propagation model as specified in [68] as follows:

$$PL_{indoor}[\text{dBm}] = 128.1 + (37 : 6 * \log_{10}(\text{D}^*0 : 001)) \tag{7}$$

$$PL_{outdoor}[\text{dBm}] = 128.1 + (37 : 6 * \log_{10}(\text{D}^*0 : 001)) + \text{L}_{wall} \tag{8}$$

where $L_{wall}$ is the external Wall Attenuation (20 dBm), if the user is in the outdoors, and D is the distance between the $MUE_i/FUE_z$ toward $F_l$.

In the first part of the examinations, the evaluation and its performance results of the power allocation scheme have been carried out through a wide range of simulations. Regarding the simulation reference which has the possibilities to choose many kinds of scheduling modules, this study used Modified Largest Weighted Delay strategy for all type application flow. All macrocell UEs and its holding HeNodeB users support two type applications; Video and VoIP in full load scenario. The Video service is a 128 kbps data source with H.264 coding [69].

This traffic is a trace-based application that sends packets according to the available realistic video trace files. Accordingly, the VoIP flows create G.729 Voice considering an OFF/ON models, in which OFF and ON periods are exponentially truncated (upper limit of 6.9 s) and distributed with mean value 3s, respectively [70]. For the OFF period, the source does not send

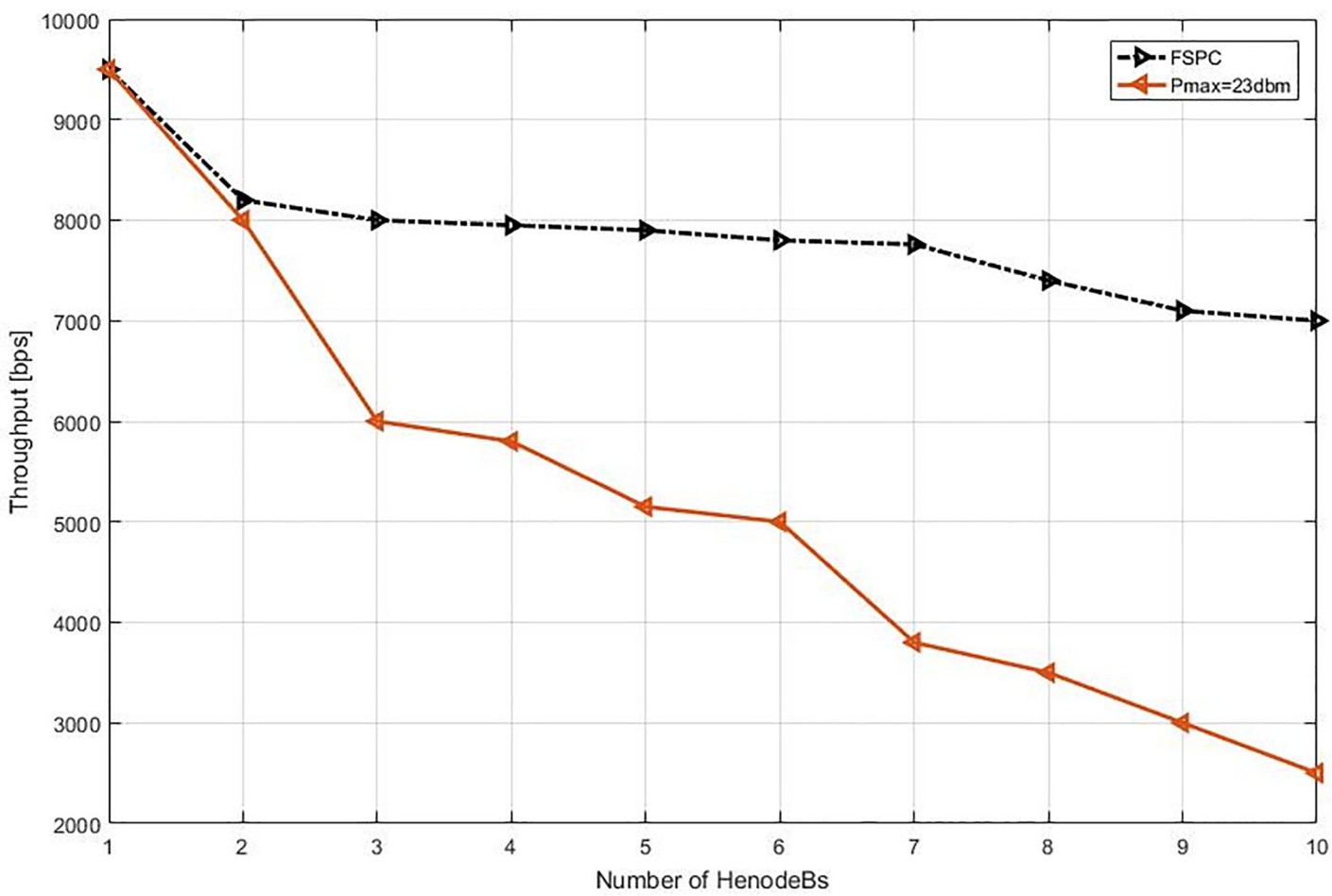

**Fig 4. Throughput average per VoIP flow (cell center).**

any sized packets due to the Voice Activity Detector while 20 bytes sized packets will be sent every 20 ms during the ON period which is equivalent to the source data rate of 8.40 kbps. In addition, the propagation loss model, namely 3GPP LTE comprises of four models (multipath, shadowing, path loss and penetration loss). Quality of Service in terms of throughput is measured for the proposed power allocation strategy FSPC are compared with maximum transmission power scheme $P_{Max}$.

## 6.2. Evaluation results

**6.2.1. HeNodeB deployment close to the eNodeB (cell center).**   In this scenario, the simulation result shows the spectral efficiency in terms throughput. While each figure display two curves.

This study specified the reliability of the strategy by showing the simulation results. For that purpose, two curves have been shown in the in the following figures (Figs 3–9). The curve made with red solid line corresponds to the maximum the transmission power value (23dbm). The curve drew with black dash line corresponds to the proposed approach which is names FSPC where the FUE UL transmitting power is changing dynamically according to the proposed strategy.

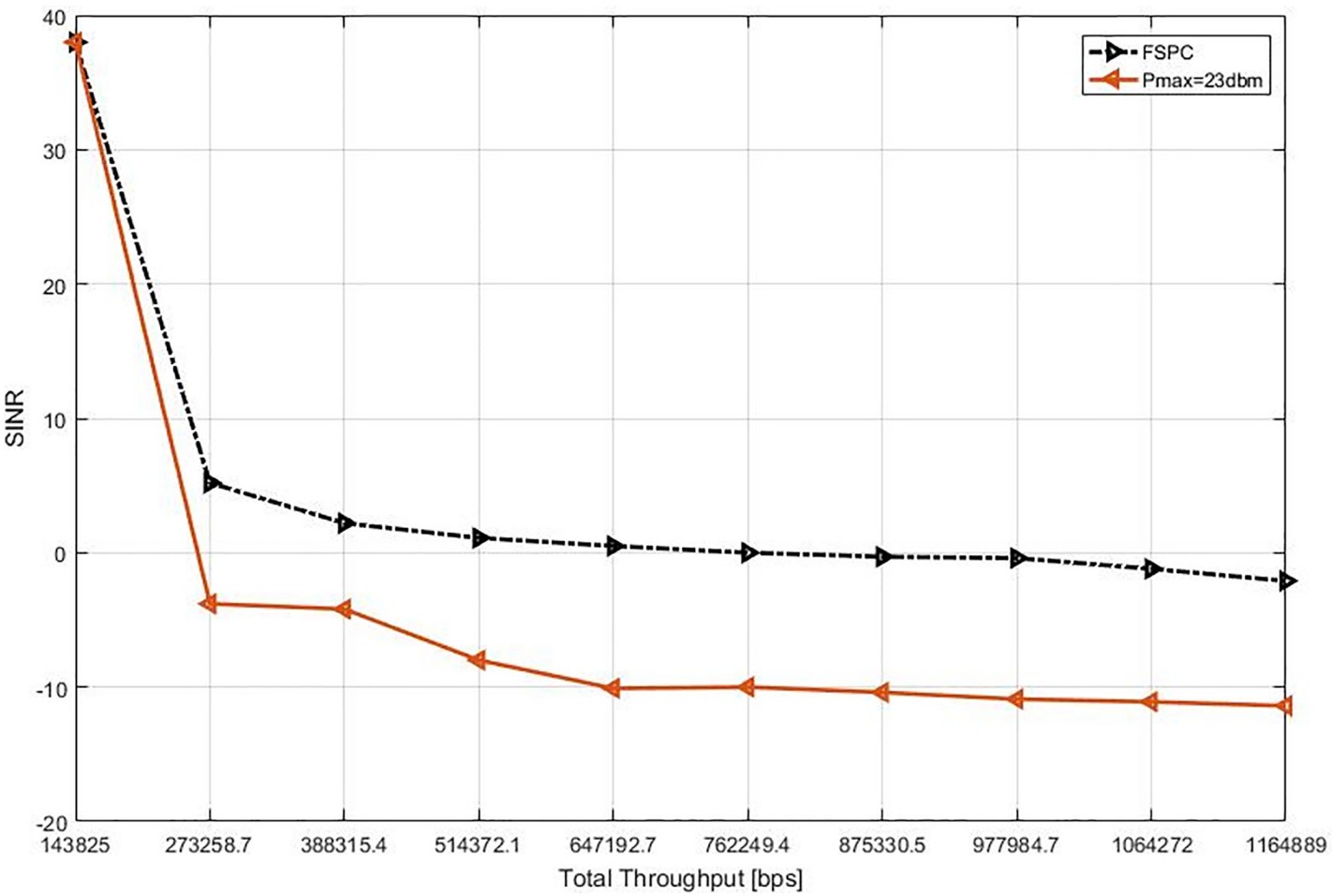

**Fig 5. Interference in video flows scenario (cell center).**

The average throughput per Video & VoIP flows are shown in Figs 3 and 4, respectively. As it can be noted that, there is slightly reduction of throughput when using FSPC approach against to the classical fixed power (Pmax = 23 dbm). However, the fixed power 23 dbm induces a high interference that leads decrease the performance preventing a favorable QoS level especially when more than two HeNodeB receive the allocated RBs at the same time. Nevertheless, when using FSPC approach, it was achieved to keep the throughput up to six HeNodeBs without a significant degradation assigning resources for video flows simultaneously. Next, The VoIP flows FSPC technique sustains the suitable throughput up to seven HeNodeBs assigning the RBs simultaneously. Furthermore, the Pmax value of small cell uplink is from 16–23 dBm.

Also, as it can be noted, the macro cell interference was not present by comparing Figs 3 and 8 of the next scenario show that the interference level slightly reduced of 1.2 dB on average. As a result of this decrement in the interference, the UL received power at HeNodeBs might be enforced. The figures also show that the average Pmax of the small cells are not excited 16.07 dBm equivalent to the same interference level as compared to the reference cases with Pmax of 23dBm in all femtocell cell deployments. This consideration guarantee that the received SINR is enhanced because of the dropped interference from each the interfering FUEs and MUEs in this deployment scenario [16, 71].

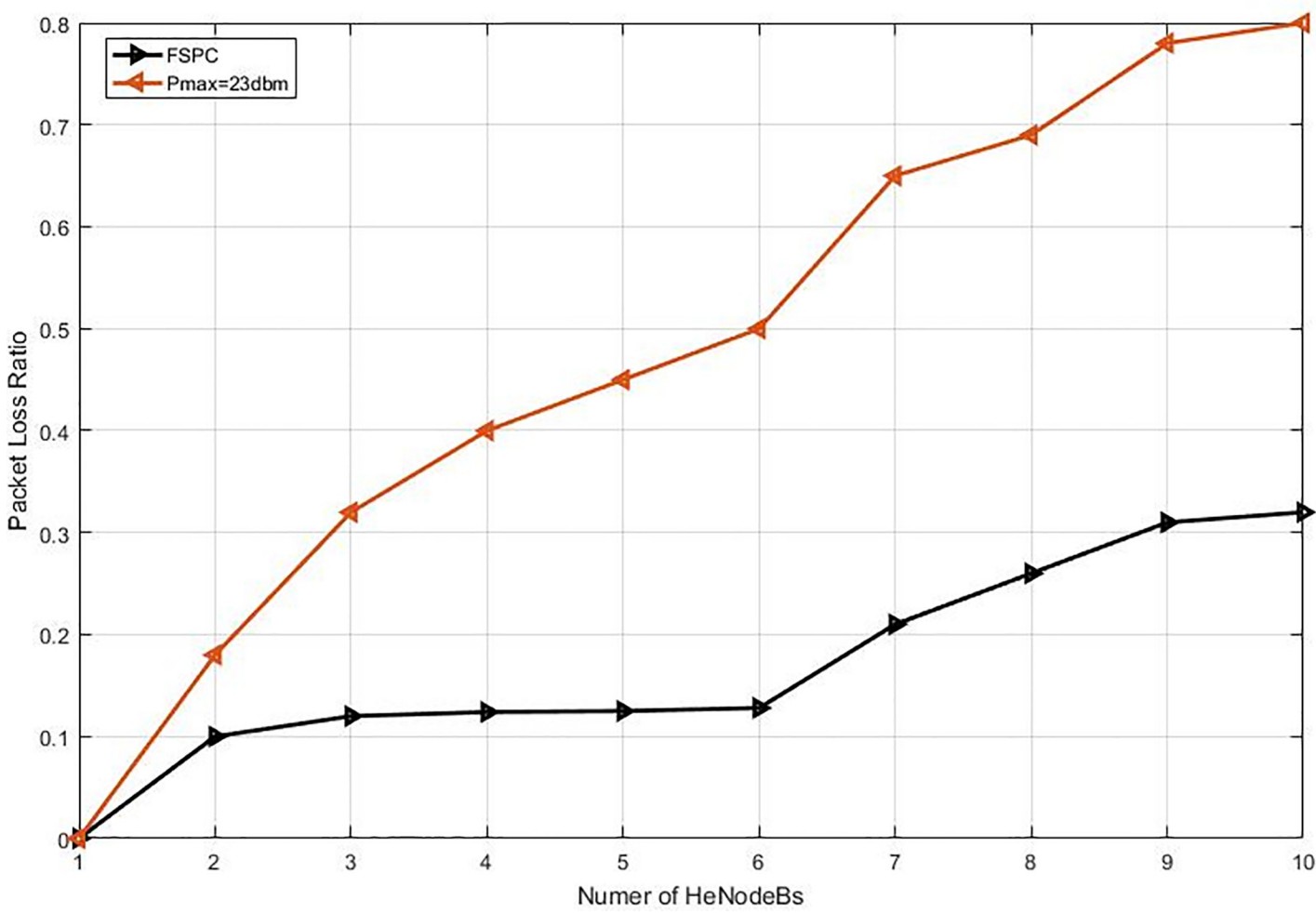

**Fig 6. Packet loss ratio for video flows (cell center).**

To view the consequences of the interference level award to the central FUEs from the interfering MUEs and-or neighboring FUEs, Fig 5 depicts interference level evaluated in terms SINR towards throughput gain for the Video flows. As indicated the interference level reduces when applying the FSPC approach over the static Pmax. It is also considered that due to the absence interfering MUEs toward FUEs, the interference level given to the FUES cell center decreased [24–27, 31]. The reason for this is that, all the MUEs around any neighboring HenodeB, which was at the distance R from the central macrocell base station, where the MUEs UL direction doesn't need much energy to transmit their allocated RBs.

Figs 6 and 7 illustrate the packet loss ratio PLR of Video and VoIP flows respectively. It can be seen from each figure that the PLS increases when the number of HeNodeB increased. This is because the number of uploading FUEs is highly correlated to the generating interference [31]. In other side, it can observe the gain in the proposed strategy is higher performance than the static strategy, this can remark that the proposed scheme has better performance in rural territory where involved a higher HeNodeBs. Accordingly, it's evaluated; the performance gain of the proposed strategy is more than 3.5% and 1.2% of the Video and VoIP. The above facts, specifically (the SINR versus the distance of MUEs as a function of impact interference in the HetNets) has been also investigated by a study of Haider, Sinha [31]. Nevertheless, the

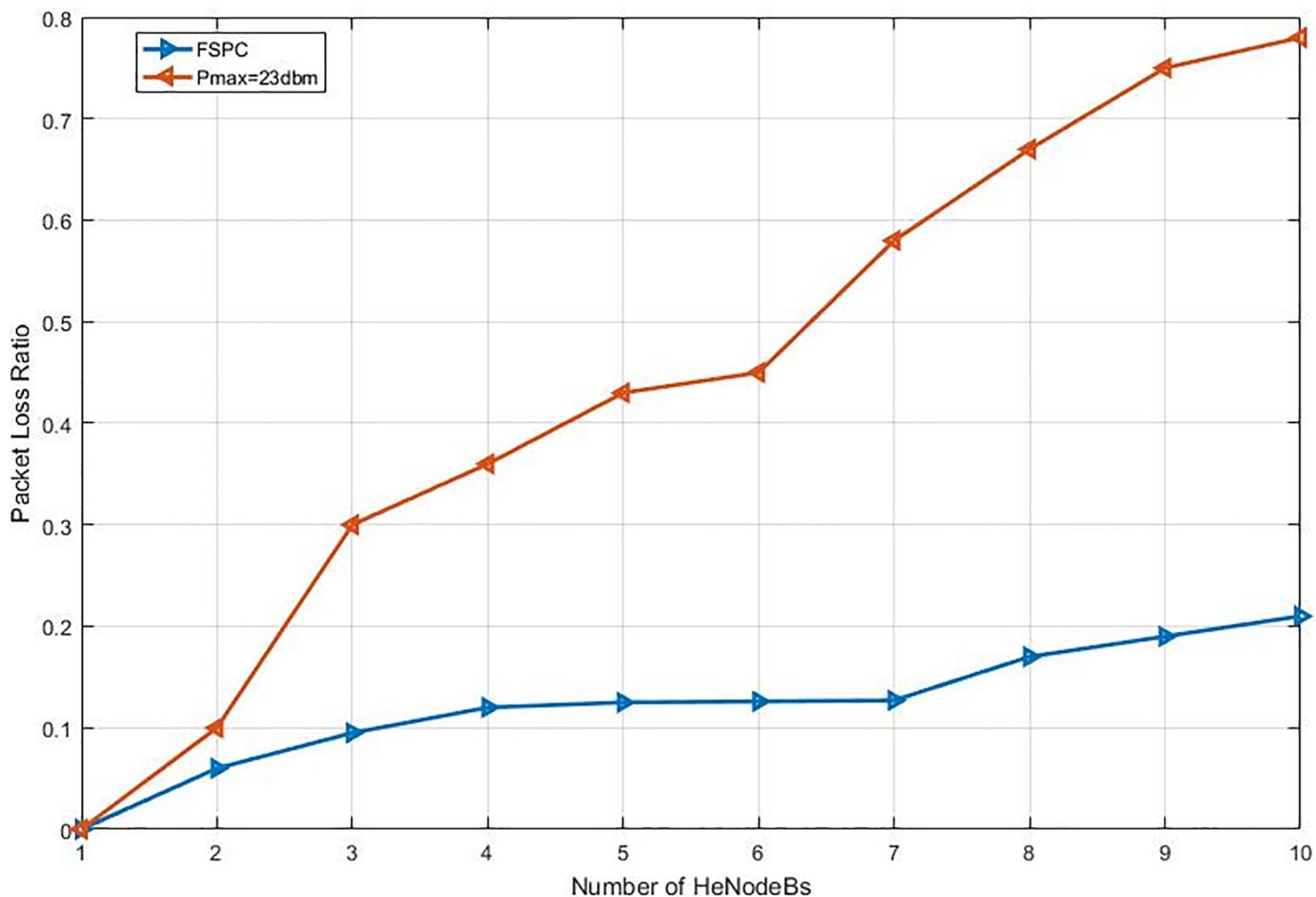

**Fig 7. Packet loss ratio for VoIP flows (cell center).**

mentioned study has not focused on a real scenario such as the motilities of the UEs either eNodeB/HeNodeB, as well as there is no detail on the way how the UEs have been applied to the selected service applications such as VoIP or Video. While the amount of power could be dynamically changed according to the MCS of each UE service. A study of Alsafasfeh, Saraereh [72] worked on the efficient power control framework for small-cell heterogeneous networks, and developed an optimization framework that allows the small-cell base station to switch off when the traffic demands are not critical. However, this might not be a practical solution.

**6.2.2. Heterogeneous network deployment with small cells near macro cell edge.** The second part of the experiment and evaluation process shows the effect of the strategic schemes on FUEs throughput near to cell edge macrocell base station. Figs 8 and 9 illustrate femtocells' average throughput for Video and VoIP, respectively, when varying the number of active FBSs. As it can be seen, the FSPC has slightly affected FUEs, the basic scenario offers, as expected, a lower average throughput for FUEs. However, it can be observed that the throughput improvement is about improvement of more than 4.5% and 2.2% in regards to Pmax. Finally, since the FSPC strategy in its design is more aware of the RT throughput degradation of FUEs, it outperforms the static regardless the traffic class.

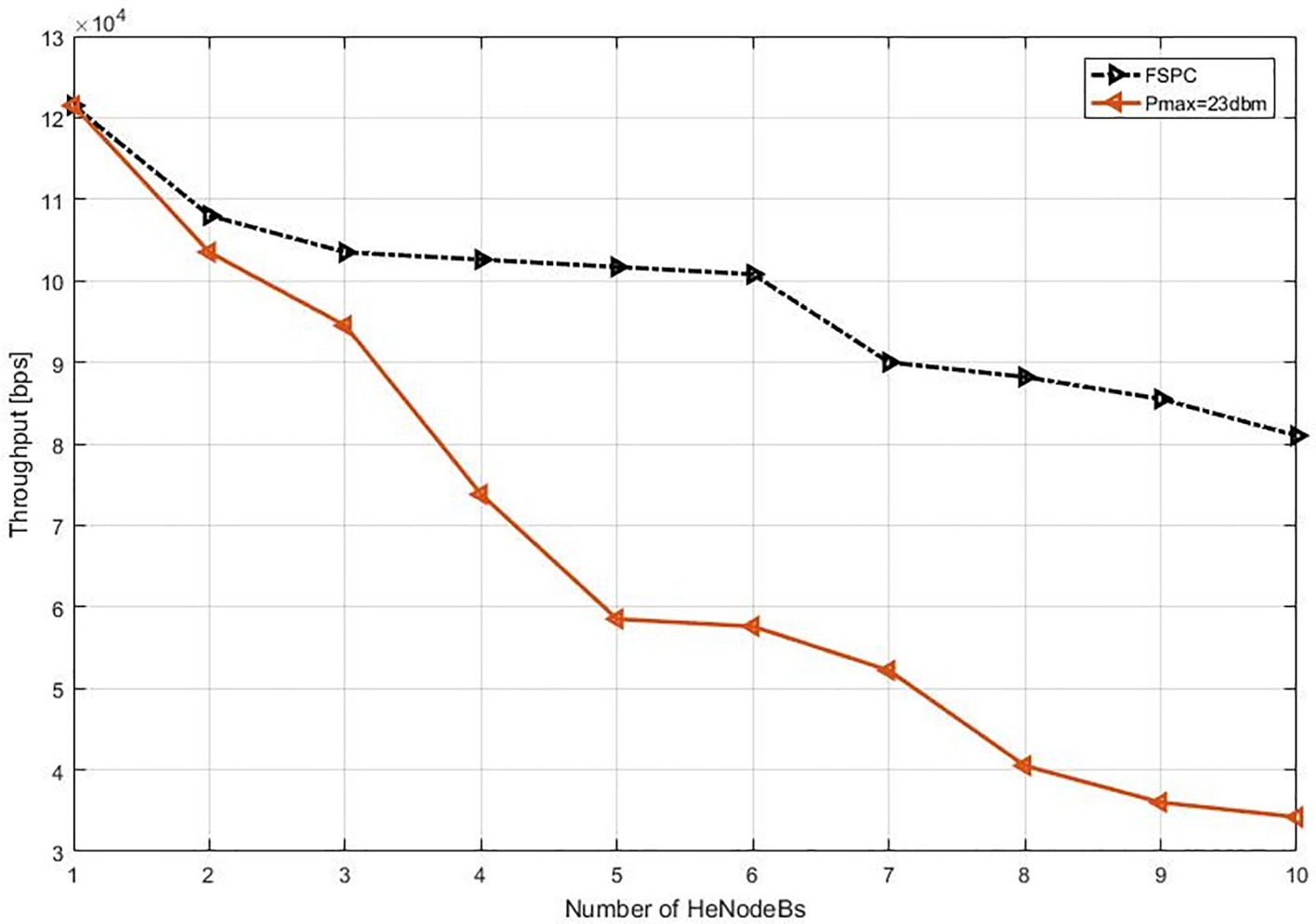

**Fig 8. Throughput average per video flow (cell edge).**

## 7. Conclusion

This study has proposed a mechanism for interference mitigation in LTE-A uplink system. HetNet deployment scenario was investigated in three performance metrics called SINR, throughput and PLR. The proposed strategy showed the effective performance of the system in utilizing RT services. This context was proved in the simulation results regarding to the controlling transmission power that is proposed in two consequences levels, where the UL transmission power value was partially adapted (decreased/increased) depending on the interference level. In this strategy, the uplink receiver at the eNodeB estimates the SINR of the received signal and compares it with the desired SINR target value according to RTT. Based on the mentioned parameters, this study named the strategy "FSPC". In the proposed strategy, TPC command is transmitted to the UE to increase the power transmitter power when the received SINR did not satisfy the SINR target. If not, the TPC command will be requested to decrease a transmitter power. In other words, FSPC specifies the minimum transmission for each UE in the first step. Then, the initial power will be further adjusted using PDCCH signaling as a TPC commands to the UE. Furthermore, in this HetNets environment FUEs UL was investigated to find optimum energy consumption in order to mitigate the interference and

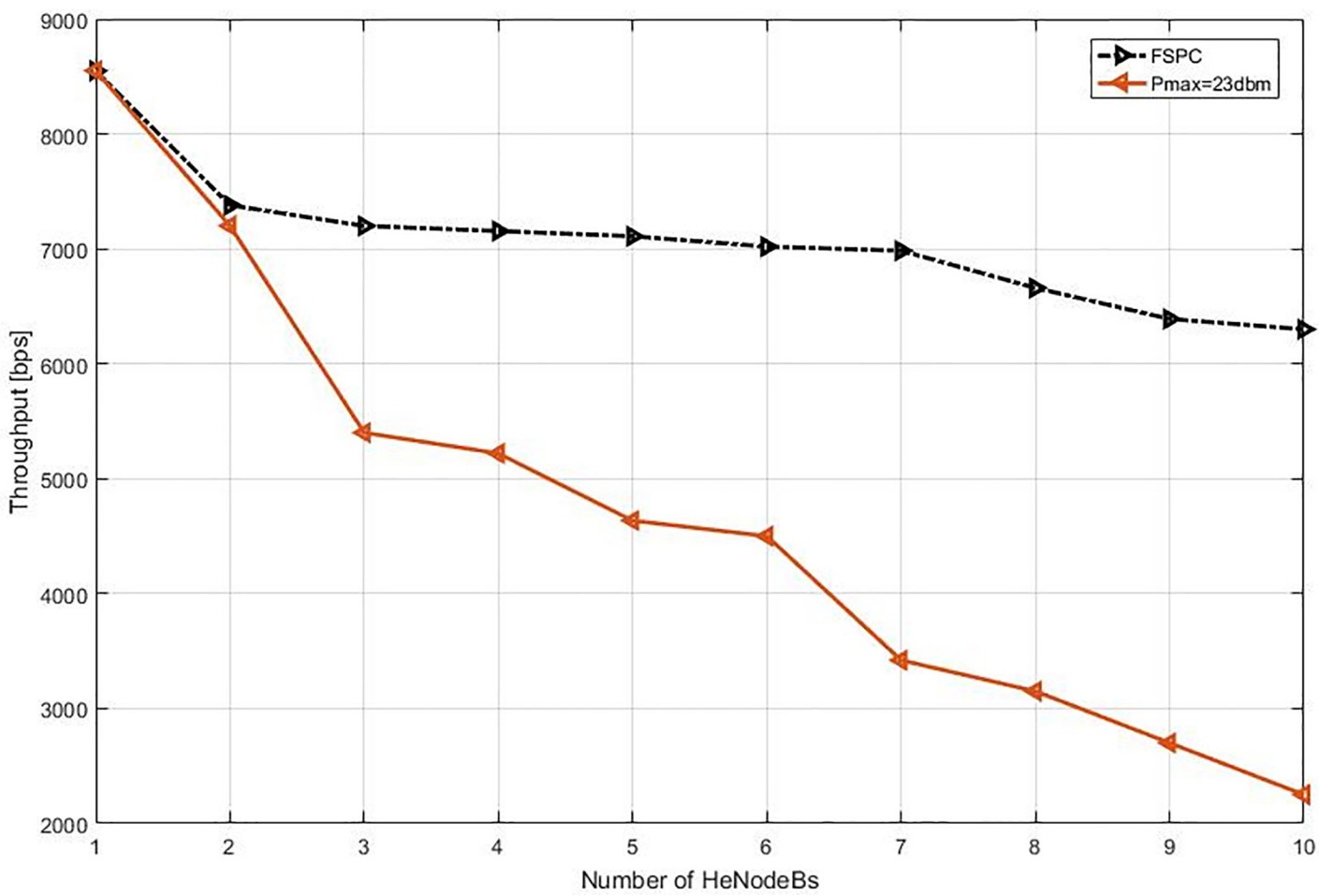

**Fig 9. Throughput average per VoIP flow (cell edge).**

well as satisfying the QoS when mobile carrying out TR services. Future work can be extended to perform MUEs performance which was neglected in the simulation scenario.

## Author Contributions

**Conceptualization:** Reben Kurda.

**Data curation:** Reben Kurda.

**Formal analysis:** Reben Kurda.

**Investigation:** Reben Kurda.

**Methodology:** Reben Kurda.

**Project administration:** Reben Kurda.

**Resources:** Reben Kurda.

**Software:** Reben Kurda.

**Supervision:** Reben Kurda.

**Validation:** Reben Kurda.

**Visualization:** Reben Kurda.

**Writing – original draft:** Reben Kurda.

**Writing – review & editing:** Reben Kurda.

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
