## [Decision Letter · Decision Letter 0]

6 Apr 2021

PONE-D-21-04508

Heterogeneous Networks: Fair Power Utilization in LTE-A Uplink scenarios

PLOS ONE

Dear Dr. KURDA,

Thank you for submitting your manuscript to PLOS ONE. After careful consideration, we feel that it has merit but does not fully meet PLOS ONE’s publication criteria as it currently stands. Therefore, we invite you to submit a revised version of the manuscript that addresses the points raised during the review process.

We look forward to receiving your revised manuscript.

Kind regards,

Anandakumar Haldorai, PhD

Academic Editor

PLOS ONE

Additional Editor Comments:

Following are the reviews for your paper referenced above. I invite you to revise and resubmit your paper within 30 days (independent of what the system shows). Please carefully address the issues raised in the comments and, up front in your revised paper, describe how the comments of each reviewer are addressed. Your revised paper will be sent to the same reviewers, as well as possibly new reviewers, for evaluation.

You may ignore any suggestion of including self-references by reviewers if not applicable.

It is recommended to use a professional native English-speaking editor. Papers with less than excellent English will not be published even if technically perfect.

Include a paragraph at the end of the Introduction describing the organization of the paper.

Journal Requirements:

Reviewers' comments:

Reviewer's Responses to Questions

**Comments to the Author**

1. Is the manuscript technically sound, and do the data support the conclusions?

Reviewer #1: Yes

Reviewer #2: Yes

2. Has the statistical analysis been performed appropriately and rigorously? 

Reviewer #1: No

Reviewer #2: Yes

3. Have the authors made all data underlying the findings in their manuscript fully available?

Reviewer #1: Yes

Reviewer #2: Yes

4. Is the manuscript presented in an intelligible fashion and written in standard English?

Reviewer #1: Yes

Reviewer #2: Yes

5. Review Comments to the Author

Reviewer #1: The authors study the problem off interference mitigation within long term evolution networks by providing initially a review analysis of the existing works that studied the impact of power control techniques in the interference mitigation regarding the network performance, and then the authors introduce an interference mitigation model in order to improve the operation of the LTE networks focusing on the uplink communication.

The authors have provided a detailed review of some existing methods studying the power control problem in the uplink of LTE networks and they have tried to categorize those approaches. Then, they introduce a femtocell-based architecture where they deal with a power control problem aiming at the interference mitigation.

They have introduced an algorithmic approach in order to deal with the power control problem and they examined several topologies in order to evaluate the proposed framework in terms of achievable throughput and the corresponding signal to interference plus noise ratio. The authors should consider the following suggestions provided by the reviewer in order to improve the quality of presentation of their manuscript, as well as its scientific depth.

Initially, in Section 2, the authors should discuss the existing resource management approaches, such as Tsiropoulou, E. E., et al. "Uplink resource allocation in SC-FDMA wireless networks: A survey and taxonomy." Computer Networks 96 (2016): 1-28, exploiting the single carrier frequency division multiple access technique in long-term evolution networks in order to deal with the problem of interference mitigation and power control.

Furthermore, the problem of power control and interference mitigation is jointly examined with the problem of bandwidth allocation, such as Tsiropoulou, E. E., et al. "Energy-efficient subcarrier allocation in SC-FDMA wireless networks based on multilateral model of bargaining." 2013 IFIP Networking Conference. IEEE, 2013, Hyung G. Myung, Junsung Lim, David J. Goodman, “Single carrier FDMA for uplink wireless transmission,”IEEE Vehicular Technology Magazine, Vol. 1, Issue 3, pp. 30 – 38, Feb. 2007, in order to provide realistic and holistic solutions that exploit the available resources of the network . The authors should improve the provided survey study in Section 2 in order to identify the existing approaches and better clarify their main contributions.

In Section 3, the authors seem to adopt a code division multiple access technique, which is relatively outdated in long term evolution networks. The authors should better clarify the choice of multiple access technique in order to perform their analysis.

In section 6, the authors should include some additional numerical results for a large scale set up with a large number of connected nodes in order to show the scalability, efficiency, and robustness of the proposed framework. This discussion should be complemented with some additional numerical simulations quantifying the computational complexity of the proposed framework. Finally, the overall manuscript should be checked for typos, syntax, and grammar errors in order to improve the quality of its presentation.

Reviewer #2: Comments:

In general, authors perform a detailed analysis on the targeted topic. The work looks solid. However, I have some comments that authors may concern:

1) the paper is entitled by "Heterogeneous Networks: Fair Power Utilization in LTE-A Uplink scenarios". however, in your manuscript, power utilization is not mentioned insufficiently. For example, in fig 2, only section mentions general power problems.

2) I suggest authors may further provide the concepts of Generation Partnership Project. In my opinion, traditional multi-layer networks cannot be regarded as Uplink (UL) direction. Authors should clarify these concepts.

3) authors may further mention the heterogeneous networks, such as done in zhang, et al. Nature-Inspired Self-Organization, Control, and Optimization in Heterogeneous Wireless Networks, ieee tmc, 2012. Authors may also investigate other types of such networks.

4) Tabulate the major findings from your previous analysis. There is no proper explanation under section 2.

5) Author may clear the contributions with case study approach.

6) Results must be properly discussed under the discussion section. There is no result comparison or analysis identified in this paper.

7) Add some new research and avoid repeated concepts in your manuscript.

Technical comments

1. The study presents the results of original research.

Yes satisfied

2. Results reported have not been published elsewhere.

Yes verified and recommended

3. Experiments, statistics, and other analyses are performed to a high technical standard and are described in sufficient detail.

The above section need some major revision, details addressed in author section

4. Conclusions are presented in an appropriate fashion and are supported by the data.

Satisfied

5. The article is presented in an intelligible fashion and is written in standard English.

The complete proofread recommended with native correction

6. The research meets all applicable standards for the ethics of experimentation and research integrity.

Maybe improved

7. The article adheres to appropriate reporting guidelines and community standards for data availability.

Yes. Available

6. PLOS authors have the option to publish the peer review history of their article (what does this mean?). If published, this will include your full peer review and any attached files.

Reviewer #1: No

Reviewer #2: No

---

## [Author Response · Author response to Decision Letter 0]

2 May 2021

Dear reviewers,

Enclosed please see the revised version of the manuscript and the reviewer's response file.

I believe all of your comments have been carefully considered. Due to these valuable comments, the quality of the manuscript is now significantly improved. I will gladly accept any other comments from you. Once again, thank you very much for your significant efforts and valuable time.

---

## [Decision Letter · Decision Letter 1]

17 May 2021

Heterogeneous Networks: Fair Power Allocation in LTE-A Uplink Scenarios

PONE-D-21-04508R1

Dear Dr. KURDA,

We’re pleased to inform you that your manuscript has been judged scientifically suitable for publication and will be formally accepted for publication once it meets all outstanding technical requirements.

Kind regards,

Anandakumar Haldorai, PhD

Academic Editor

PLOS ONE

Additional Editor Comments (optional):

The authors have addressed all the issues raised.

Recommended for further publication process.

Reviewers' comments:

Reviewer's Responses to Questions

**Comments to the Author**

1. If the authors have adequately addressed your comments raised in a previous round of review and you feel that this manuscript is now acceptable for publication, you may indicate that here to bypass the “Comments to the Author” section, enter your conflict of interest statement in the “Confidential to Editor” section, and submit your "Accept" recommendation.

Reviewer #1: All comments have been addressed

Reviewer #2: All comments have been addressed

2. Is the manuscript technically sound, and do the data support the conclusions?

Reviewer #1: Yes

Reviewer #2: Yes

3. Has the statistical analysis been performed appropriately and rigorously? 

Reviewer #1: Yes

Reviewer #2: Yes

4. Have the authors made all data underlying the findings in their manuscript fully available?

Reviewer #1: Yes

Reviewer #2: Yes

5. Is the manuscript presented in an intelligible fashion and written in standard English?

Reviewer #1: Yes

Reviewer #2: Yes

6. Review Comments to the Author

Reviewer #1: The authors have addressed all the reviewers comments. The quality of presentation and the scientific depth of the manuscript have been substantially improved.

Reviewer #2: The paper discussed about Fair Power Utilization in Small Cell Uplink Scenarios in Heterogeneous Networks

The revised version of paper has improved.

7. PLOS authors have the option to publish the peer review history of their article (what does this mean?). If published, this will include your full peer review and any attached files.

Reviewer #1: No

Reviewer #2: No

---

## [Editor Report · Acceptance letter]

25 May 2021

PONE-D-21-04508R1 

Heterogeneous Networks: Fair Power Allocation in LTE-A Uplink Scenarios 

Dear Dr. Kurda:

I'm pleased to inform you that your manuscript has been deemed suitable for publication in PLOS ONE. Congratulations! Your manuscript is now with our production department. 

Kind regards, 

on behalf of

Dr. Anandakumar Haldorai 

Academic Editor

PLOS ONE